# Thermomass Theory in the Framework of GENERIC

**DOI:** 10.3390/e22020227

**Published:** 2020-02-18

**Authors:** Ben-Dian Nie, Bing-Yang Cao, Zeng-Yuan Guo, Yu-Chao Hua

**Affiliations:** Key Laboratory for Thermal Science and Power Engineering of Ministry of Education, Department of Engineering Mechanics, Tsinghua University, Beijing 100084, China; nbd15@mails.tsinghua.edu.cn (B.-D.N.); caoby@tsinghua.edu.cn (B.-Y.C.); demgzy@tsinghua.edu.cn (Z.-Y.G.)

**Keywords:** GENERIC, thermomass, thermomass energy, hyperbolic heat conduction

## Abstract

Thermomass theory was developed to deal with the non-Fourier heat conduction phenomena involving the influence of heat inertia. However, its structure, derived from an analogy to fluid mechanics, requires further mathematical verification. In this paper, General Equation for Non-Equilibrium Reversible-Irreversible Coupling (GENERIC) framework, which is a geometrical and mathematical structure in nonequilibrium thermodynamics, was employed to verify the thermomass theory. At first, the thermomass theory was introduced briefly; then, the GENERIC framework was applied in the thermomass gas system with state variables, thermomass gas density *ρ*_h_ and thermomass momentum **m**_h_, and the time evolution equations obtained from GENERIC framework were compared with those in thermomass theory. It was demonstrated that the equations generated by GENERIC theory were the same as the continuity and momentum equations in thermomass theory with proper potentials and eta-function. Thermomass theory gives a physical interpretation to the GENERIC theory in non-Fourier heat conduction phenomena. By combining these two theories, it was found that the Hamiltonian energy in reversible process and the dissipation potential in irreversible process could be unified into one formulation, i.e., the thermomass energy. Furthermore, via the framework of GENERIC, thermomass theory could be extended to involve more state variables, such as internal source term and distortion matrix term. Numerical simulations investigated the influences of the convective term and distortion matrix term in the equations. It was found that the convective term changed the shape of thermal energy distribution and enhanced the spreading behaviors of thermal energy. The distortion matrix implies the elasticity and viscosity of the thermomass gas.

## 1. Introduction

Fourier’s law, which implied a parabolic governing equation, failed to describe heat conduction phenomena in ultrafast processes [1,2,3,4,5,6,7] and nanoscale materials [8,9,10,11,12,13,14]. For instance, heat perturbation propagated in the medium at a finite speed, according to the observation in heat pulse experiments, while the parabolic equation predicted a paradox with infinite speed [7]. In nanoscale materials, the thermal conductivity was found to be size-dependent, which was beyond the scope of Fourier’s law [13].

In order to characterize non-Fourier heat conduction, various models had been proposed based on different assumptions and physical pictures. Hyperbolic heat conduction equations were preferred because they solved the paradox of infinite thermal disturbance speed [15] and, in recent studies, investigated the phonon momentum conservation process [16,17], which implied the inertial phenomena of heat flux [18,19]. Cattaneo-Vernotte equation [20,21], the first mathematical hyperbolic heat conduction equation, assumed a relaxation process between temperature gradient and heat flux, giving a telegraph equation when coupled with the energy conservation equation. However, the equation predicted negative temperature at some cases which is unphysical [22]. Afterwards, Phase lag models, such as Dual Phase Lag model [23,24], were proposed and thought that a time lag existed between heat flux and temperature gradient and it could reduce to Cattaneo-Vernotte equation via expansions. However, it was found that the solutions were not continually dependent on the initial condition [19], which resulted in instability, and it could not avoid the problem of negative temperature [25,26,27]. Another series of non-Fourier equations were derived from the Boltzmann equation [28,29,30]. Guyer-Krumhansl equation [28,29], typical for this kinds of equations led to the investigations into phonon hydrodynamics, which was named due to its similar structure with Navier-Stokes equation. However, it also met mathematical difficulties [31,32], such as the violation of maximum principle for stability and the failure in preserving variable non-negativity. Recently, to study the heat transfer in a more intrinsic way, thermomass theory [33,34,35,36] was developed. It was found that the heat was conserved during conduction, similar to mass. Then, the mass nature of *heat* was recognized in heat conduction, which is called thermomass and defined based on Einstein’s mass-energy relation. Consequently, the heat conduction is actually the thermomass gas flow through the porous medium with the rest mass as skeleton. Thus, heat conduction problems can be described by use of the principles in fluid mechanics. Thermomass theory was adopted to describe the non-Fourier heat conduction phenomena, such as hyperbolic heat conduction [22,34], negative entropy production problem [37,38] and thermomass gas viscosity [39]. Nevertheless, its mathematical structure, derived from analogy, was supposed to be testified.

Nonequilibrium thermodynamics paved another way to handle this issue, and it tried to understand the non-Fourier phenomena in the view of entropy evolution [40]. Classical Irreversible Thermodynamics (CIT) faced challenges when the parabolic Fourier’s law was extended to the hyperbolic heat conduction. The oscillations of heat flux and temperature led to the negative entropy production, which violated the Second Law. To solve this problem, CIT was extended mainly in three paths [41]. The first was to construct the conservation equations along with the time evolution equation of entropy, such as the Extended Irreversible Thermodynamics (EIT) [42,43]. The second path was to add new internal variables and get their evolution equations along with that of entropy, such as Weakly Nonlocal Irreversible Thermodynamics [44,45,46], Rational Extended Thermodynamics [47] and Conservation Dissipation Formulism [48]. The last path was to obtain an abstract geometrical structure extracted from the classical equilibrium thermodynamics. Following this, the General Equation for Non-Equilibrium Reversible-Irreversible Coupling theory [49,50,51] (abbreviated to GENERIC) was developed.

The GENERIC theory intends to provide a unified geometrical and mathematical structure for various processes, such as complex fluids [52], chemical reaction [53] and Fourier and non-Fourier heat conduction [54], and tries to demonstrate the similar structure shared by them. One of its advantages is analyzing the reversible process and irreversible process individually. The reversible process is described by the Hamiltonian mechanics, governed by the Hamiltonian energy. Moreover, the irreversible process adopts dissipation potential, which reaches its minimum when the system arrives at equilibrium state. The time evolution equations of state variables can be derived from the geometry:(1)$$\frac{dn}{dt}=LH_{n}+\frac{\partial \Xi }{\partial \eta_{n}},$$
which involves five building blocks: the state variables *n*, their Hamiltonian kinematics *L*, Hamiltonian energy *H*, dissipation potential Ξ and entropy *η*, which is also called the eta-function. The first term in the right represents the reversible process while the second term in the right is characterized by irreversibility. As for heat conduction [54], it is natural to choose energy density *e* and heat flux *q* as the state variables. The heat flux is treated in a dynamical way, but the physical interpretation is unclear, especially for the Hamiltonian potential. Therefore, the analysis of heat conduction in GENERIC structure is mainly mathematical, lacking some physical interpretation.

This paper aims to apply the thermomass theory in the framework of GENERIC, demonstrating the mathematical reasonability of the governing equations of thermomass gas and the physical foundation of the heat conduction in the framework of GENERIC.

## 2. Thermomass Theory

The part will introduce the thermomass theory briefly. Heat is conserved in heat conduction, which indicates the mass nature of heat. Therefore, it is also called thermomass. The transport process of heat can be regarded as the flowing behaviors of thermomass gas in porous media. Thus, in this way, the heat conduction phenomena can be described by fluid mechanics. Guo et al. [36] noticed these characteristics and proposed the thermomass theory, which focused on the extra mass increased due to thermal vibration. The theory aims to develop the dynamics of thermomass gas in heat conduction. In fact, Grmela [55] shared the same idea and regarded the hyperbolic heat conduction in fluid as the transport of two component fluid in his paper in 1983. The first component was fluid particle and the second was called caloric component.

The relativistic mass of an object is defined based on the Einstein’s mass-energy relation:(2)$$m=\frac{E}{c^{2}},$$
where *E* is the mechanical energy and *c* is light speed. Similar to the mechanical energy, the relativistic mass of thermal energy in a dielectric solid is:(3)$$m_{h}=\frac{E_{h}}{c^{2}}$$
which is called thermomass, as extra mass added to the rest mass. As for heat conduction in the dielectric solid, thermomass is transported, while the molecules stay in the original places. 

The dynamics of thermomass gas is based on two state variables, *ρ_h_* and **u***_h_*, representing the local thermomass gas density and drift velocity of thermomass gas in per unit volume, respectively, which are defined as:(4)$$\rho_{h}=\frac{\rho C_{V}T}{c^{2}},$$
(5)$$u_{h}=\frac{q}{\rho C_{V}T}.$$
where *ρ* is the material density and *ρ_h_* is the density of thermomass gas. *C_V_* is the specific heat. *T* and **q** are the local temperature and heat flux, respectively. Then, three conservation laws are established, namely the thermomass continuity equation, the thermomass momentum equation and the thermomass energy equation:(6)$$\frac{\partial \rho_{h}}{\partial t}+\nabla \cdot (u_{h}\rho_{h})=0,$$
(7)$$\frac{\partial (\rho_{h}u_{h})}{\partial t}+\nabla \cdot (\rho_{h}u_{h}u_{h})=-\nabla P_{h}+f_{h},$$
(8)$$\frac{\partial (\frac{1}{2}\rho_{h}u_{h}^{2})}{\partial t}+\nabla \cdot (u_{h}\cdot \frac{1}{2}\rho_{h}u_{h}^{2})=-\nabla P_{h}\cdot u_{h}+f_{h}\cdot u_{h}.$$
where **u***_h_***u***_h_* in Equation (7) is a second order tensor and **u***_h_*·**u***_h_* is a scalar. The first equation corresponds to the heat conservation equation, which is:(9)$$\frac{\partial (\rho C_{V}T)}{\partial t}+\nabla \cdot q=0.$$

In the thermomass momentum Equation (7), the first term represents the temporal inertial force of thermomass while the second term represents the spatial inertial force. *P_h_* is the thermomass gas pressure defined as a quadratic function [33]:(10)$$P_{h}=\frac{\gamma_{h}\rho }{c^{2}}{\left(C_{V}T\right)}^{2},$$
and **f***_h_* is the resistance, usually assumed to have a linear relationship with the drift velocity of thermomass gas:(11)$$f_{h}=-\mu u_{h}.$$
The coefficient *μ* can be derived from the comparisons with Fourier’s law. Of course, different resistance models lead to different time evolution equations. For example, Dong et al. [39] adopted Brinkman’s equation for the resistance force to describe the boundary influences on thermomass propagation in microscale medium. When the temporal inertial force and spatial inertial force can be ignored, the thermal pressure is equal to the thermal resistance [36], namely:(12)$$\nabla P_{h}=f_{h}.$$
At that time, Equation (12) is actually Fourier’s law, so that *μ* can be expressed in the form of thermal conductivity *k* as:(13)$$\mu =\frac{2c^{2}\gamma_{h}\rho_{h}^{2}C_{V}}{k}.$$
Then, the thermomass momentum equation (Equation (7)) becomes:(14)$$\frac{\partial q}{\partial t}+\nabla \cdot (\frac{qq}{\rho C_{V}T})+\frac{2\gamma_{h}\rho C_{V}^{2}T}{c^{2}}\nabla T+\frac{2\gamma_{h}\rho C_{V}^{2}T}{c^{2}k}q=0.$$
where **qq** is a second order tenser. The time evolution equation of heat flux **q** is derived with temperature *T* on the foundation of thermomass gas dynamics. This equation can reduce to the conventional Fourier’s conduction law when all the inertia effects are omitted. It reduces to the hyperbolic heat conduction equation when the spatial inertia is neglected and it is also able to describe the non-Fourier heat conduction phenomena even in the steady state, when the temporal inertia is omitted. Therefore, Equation (14) is called the general heat conduction law in the references [35,36].

The conservation equation of thermomass energy can be derived from the coupling of thermomass continuity equation and momentum equation, while new quantities are defined as the thermomass kinetic energy (Equation (15)) and thermomass potential energy (Equation (16)):(15)$$E_{hk}=\frac{1}{2}\rho_{h}u_{h}\cdot u_{h}=\frac{q\cdot q}{2\rho C_{v}Tc^{2}},$$
(16)$$E_{hp}=\frac{\gamma_{h}\rho }{c^{2}}{\left(C_{V}T\right)}^{2}.$$
In the transport process where the drift velocity of thermomass gas is not so low that the kinetic energy of thermomass cannot be neglected, the kinetic energy and potential energy of thermomass convert into each other mutually cycle by cycle, and the system goes through the oscillations of heat flux and temperature. 

It is important to emphasize that there are two systems in a solid. The first one is the porous skeleton of rest mass of molecules and the other is the thermomass gas, which is full of the porous skeleton and can flow through it, if a temperature difference is applied to the solid. Heat conduction in a solid is in fact the flowing of thermomass gas in a porous medium, which can be described by the principles of fluid dynamics. The quantities in the porous skeleton system are those we are familiar with, such as density *ρ* and specific heat *C_V_*. In the thermomass gas system, the quantities are labeled by the subscript *h*, such as *ρ_h_* and *u_h_*. *ρ_h_* is the density of thermomass gas, which is the function of the density of rest mass skeleton, *ρ*, as shown in Equation (4). It is assumed that the propagation of thermomass gas is only influenced by the thermomass gas system, free from the skeleton system.

## 3. Thermomass Theory in the Framework of GENERIC

GENERIC theory divides the whole process into the reversible part and the irreversible part, controlled by two potentials, namely the Hamiltonian potential and the dissipation potential. Thermomass gas can be regarded as a kind of special fluid, and thus its transport process can be dealt with in the framework of GENERIC with the analogy to fluid mechanics. Because different state variables take effect in different cases, choosing proper state variables is significant, which makes the GENERIC theory easier to be extended. In the thermomass system, there are two elementary state variables to choose, thermomass gas density *ρ_h_* and thermomass gas flow **m***_h_*, which represents the thermomass gas momentum, corresponding to the thermomass gas density *ρ_h_* and thermomass gas drift velocity **u***_h_* in thermomass theory by:(17)$$m_{h}=\rho_{h}u_{h}=\frac{q}{c^{2}}.$$
In this paper, the thermomass system based on the solid system is studied. Then, the Poisson bracket are derived via Hamiltonian mechanics [51]:(18)$$\begin{array}{ll} {\left\{A,H\right\}}^{c} & =\int_{V}\rho_{h}(\frac{\partial A_{\rho_{h}}}{\partial x_{i}}H_{m_{h,i}}-\frac{\partial H_{\rho_{h}}}{\partial x_{i}}A_{m_{h,i}})dr\\ & +\int_{V}m_{h,i}(\frac{\partial A_{m_{h,i}}}{\partial x_{j}}H_{m_{h,j}}-\frac{\partial H_{m_{h,i}}}{\partial x_{j}}A_{m_{h,j}})dr \end{array}$$
where Einstein’s summation convention is used for the subscript *i*, *j* and *m_h_*_,*i*_ represents the *i*_th_ component of the vector **m***_h_*. Equation (18) is obtained from the analogy to the Poisson bracket in fluid mechanics [51]. In this equation, *A* is a function with respect to thermomass density and thermomass momentum *A*(*ρ_h_*, **m***_h_*) and represents some system properties. *H* is the Hamiltonian energy in thermomass system. Poisson bracket depicts the reversible process, including the propagation and the transition between thermomass kinetic energy and thermomass potential energy. The Poisson bracket is supposed to satisfy three basic principles, namely the linearity, {*A*, *H*} + {*B*, *H*} = {*A* + *B*, *H*}, the antisymmetry, {*A*, *H*} = −{*H*, *A*} and the Jacobi identity {{*A*, *B*}, *H*} + {{*B*, *H*}, *A*} + {{*H*, *A*}, *B*}= 0. These principles are testified for Equation (18) since its structure is similar to that in fluid mechanics. Then, the following equations are obtained via integration by parts:(19)$$\dot{A}=\frac{\partial A}{\partial \rho_{h}}\frac{\partial \rho_{h}}{\partial t}+\frac{\partial A}{\partial m_{h,i}}\frac{\partial m_{h,i}}{\partial t},$$
(20)$$\dot{A}=\{A,H\},$$
(21)$$\begin{array}{l} {\left\{A,H\right\}}^{c}=\int dV(-A_{\rho_{h}}\frac{\partial }{\partial x_{i}}(\rho_{h}H_{m_{h,i}})-A_{m_{h,i}}\rho_{h}\frac{\partial H_{\rho_{h}}}{\partial x_{i}})\\ +\int dV(-A_{m_{h,i}}\frac{\partial }{\partial x_{j}}(m_{h,i}H_{m_{h,j}})-A_{m_{h,j}}(m_{h,i}\frac{\partial H_{m_{h,i}}}{\partial x_{j}}) \end{array}$$

The time evolution equations of the reversible process can be derived by substituting Equation (19) into Equation (21):(22)$$\frac{\partial \rho_{h}}{\partial t}+\frac{\partial }{\partial x_{i}}(\rho_{h}H_{m_{h,i}})=0,$$
(23)$$\frac{\partial m_{h,i}}{\partial t}+\frac{\partial }{\partial x_{j}}(m_{h,i}H_{m_{h,j}})=-\rho_{h}\frac{\partial H_{\rho_{h}}}{\partial x_{i}}-m_{h,j}\frac{\partial H_{m_{h,j}}}{\partial x_{i}}.$$
A so-called pressure potential is introduced to simplify the expression, as:(24)$$\pi =H-\rho_{h}H_{\rho_{h}}-m_{h,j}H_{m_{h,j}}.$$
The construction of such pressure potential is common in Hamiltonian mechanics and it implies the transition between kinetic energy and potential energy. Equation (23) is transformed into:(25)$$\frac{\partial m_{h,i}}{\partial t}+\frac{\partial }{\partial x_{j}}(m_{h,i}H_{m_{h,j}})=\frac{\partial \pi }{\partial x_{i}}$$
by substituting Equation (24) into Equation (23). These Equations (22) and (23) describe a non-dissipative transport process of thermomass and thermomass momentum. They are supposed to correspond with the following hyperbolic heat conduction equations without dissipation:(26)$$\tau \frac{\partial q_{i}}{\partial t}=-k_{i}\frac{\partial T}{\partial x_{i}},$$
(27)$$\rho C_{V}\frac{\partial T}{\partial t}=-\frac{\partial q_{i}}{\partial x_{i}}.$$

It has to be clarified that the non-dissipative process described by Equations (26) and (27) does not exist in reality. They are analyzed only as the reversible part of the whole heat transport process and what occurs in the process is the combination of the reversible part and the irreversible part. Through the comparisons between Equations (26) and (27) and Equations (22) and (23), it is found that a proper definition of Hamiltonian potential *H* makes these two equation sets become the same. A specific form of *H* can be chosen as:(28)$$H_{2}=\sum_{i}\frac{1}{2}\frac{m_{h,i}^{2}(x,t)}{\rho_{h}(x,t)}+\frac{1}{2}\varepsilon_{2}\rho_{h}^{2}(x,t),$$
which is the thermomass energy in the thermomass theory, consisting of kinetic energy and potential energy. Moreover, *ε* is a constant with dimension m^5^kg^−1^s^−2^ to make sure that the Hamiltonian energy *H* has the dimension J/m^3^, which actually means the energy density in the unit volume. Taking into consideration the relationships Equations (4), (5), (28), Equation (22) and Equation (23) are transformed into:(29)$$\frac{\partial e}{\partial t}=-\frac{\partial q_{i}}{\partial x_{i}},$$
(30)$$\frac{\partial q_{i}}{\partial t}+\frac{\partial (\frac{q_{i}q_{j}}{\rho C_{V}T})}{\partial x_{j}}=-\varepsilon_{2}\frac{e}{c^{2}}\frac{\partial e}{\partial x_{i}}$$
and the parameter *ε* is defined as:(31)$$\varepsilon_{2}=\frac{c^{2}k}{(\rho C_{v})e}.$$

When the variation of temperature is not large, linear approximation can be adopted and the variation of energy density *e* in the expression (31) is omitted. In this way, the coefficient *ε*_2_*e*/*c*^2^ in the right of the Equation (30) gives a linear energy-dependent relationship. Of course, the formulations of Hamiltonian energy are not unique, a linear or a cubic relationship for thermomass potential energy fits the equations as well. However, it is demonstrated that the linear relationship:(32)$$H_{1}=\sum_{i}\frac{1}{2}\frac{m_{h,i}^{2}(x,t)}{\rho_{h}(x,t)}+\varepsilon_{1}\rho_{h}(x,t)$$
leads to a constant pressure potential:(33)$$\frac{\partial \pi_{1}}{\partial x_{i}}=0$$
which is not suitable in the physical system. The cubic relation:(34)$$H_{3}=\sum_{i}\frac{1}{2}\frac{m_{h,i}^{2}(x,t)}{\rho_{h}(x,t)}+\varepsilon_{3}\rho_{h}^{3}(x,t)$$
requires that the thermal conductivity relies on the quadratic energy density *e*^2^, which is not considered in this paper:(35)$$\frac{\partial \pi_{3}}{\partial x_{i}}=-\varepsilon_{3}\frac{e^{2}}{c^{6}}\frac{\partial e}{\partial x_{i}}.$$

As for the irreversible processes, eta-function and dissipation potential are introduced and the time evolution equations are in entropy representation. Eta-function *η*, whose derivative with respect to energy density *e*(*x*,*t*) in local equilibrium state equals the inverse temperature:(36)$${\frac{\partial \eta }{\partial e}|}_{e(x)=e_{eq}(x)}=\frac{1}{T_{eq}},$$
has the meaning of entropy. The subscript *eq* represents the equilibrium state. *η_e_*, abbreviation for the derivative of *η* with respect to *e* is called the conjugate variable of *e*. In the GENERIC theory, contact geometry and conjugate variables are widely adopted. In equilibrium state, the eta-function has the expression:(37)$$\eta {|}_{e(x)=e_{eq}(x)}=\rho C_{V}\ln T_{eq}+D.$$
in solid physics with constant *D*. According to the maximum entropy principle, the eta-function approaches its maximum during the process from nonequilibrium state to equilibrium state. *η*(*ρ_h_*, **m***_h_*) is the function of thermomass density and thermomass momentum and its variation is determined by these two state variables:(38)$$\delta \eta =\frac{\partial \eta }{\partial n}\delta n=\frac{\partial \eta }{\partial \rho_{h}}\delta \rho_{h}+\frac{\partial \eta }{\partial m_{h,i}}\delta m_{h,i}.$$

The introduction of dissipation potential in GENERIC theory extends the irreversible process from linear gradient dynamics to nonlinear cases. The time evolution can be written in the form of inner product:(39)$$\dot{A}=<A_{n},\frac{\partial \Xi }{\partial \eta_{n}}>=\frac{\partial A}{\partial n}\cdot \frac{\partial \Xi }{\partial \eta_{n}},$$
and when *A* is the eta-function itself, it is supposed to satisfy that:(40)$$\dot{\eta }=\eta_{n}\cdot \frac{\partial \Xi }{\partial \eta_{n}}>0.$$
Keeping that in mind, the derivative of *A* with respect to time could be seen as the sum of the time evolution of state variables (Equation (38)), Equation (39) could be transformed into:(41)$$\frac{dn}{dt}=\frac{\partial \Xi }{\partial \eta_{n}}.$$

As for how to get a proper dissipation potential, there are many choices. It seems that the simplest form is the quadratic one:(42)$$\Xi =\frac{\kappa }{2}{\left(\eta_{n}\right)}^{2},$$
(43)$$\frac{\partial \Xi }{\partial \eta_{n}}=\kappa (\eta_{n}).$$

One thing to be noticed is that the dissipation potential might be not only the function of state variables, but also the gradient of state variables and so on. Therefore, the specific form of dissipation potential is pertinent to the material types, microstructures and other characteristic properties. Here, one typical form can be chosen as:(44)$$\eta =\eta_{0}-\sum_{i}\frac{1}{2}\gamma_{i}m_{h,i}^{2},$$
(45)$$\Xi =\frac{1}{2}\sum_{i}\kappa_{i}\eta_{m_{h,i}}^{2}=\frac{1}{2}\sum_{i}\kappa_{i}\gamma_{i}^{2}m_{h,i}^{2}.$$
The absolute value of *η*_0_ is difficult to calculate in practice as it is related with the entropy in equilibrium state. However, luckily, only the derivatives of eta-function with respect to the state variables are necessary in the equation set. The dissipation potential is free from the conjugate variable $\eta_{\rho_{h}}$, because the thermomass is conserved, rather than being reduced during the heat conduction process.

The time evolution can be derived by combining the reversible process and irreversible process:(46)$$\frac{\partial \rho_{h}}{\partial t}+\frac{\partial }{\partial x_{i}}(\rho_{h}H_{m_{h,i}})=0,$$
(47)$$\frac{\partial m_{h,i}}{\partial t}+\frac{\partial }{\partial x_{j}}(m_{h,i}H_{m_{h,j}})=\frac{\partial \pi }{\partial x_{i}}-\kappa_{i}\gamma_{i}m_{h,i}.$$
In the nonequilibrium heat conduction process, the thermomass is transferred and conserved while the thermomass momentum is reduced until a steady state is established.

## 4. Discussion and Extensions

### 4.1. Comparisons Between GENERIC Theory and Thermomass Theory

In the thermomass theory, the momentum equation is in the form of thermomass density and drift velocity of thermomass gas:(48)$$\frac{\partial (\rho_{h}u_{h,i})}{\partial t}+\frac{\partial (\rho_{h}u_{h,j}u_{h,i})}{\partial x_{j}}=-\frac{\partial [\frac{\gamma \rho }{c^{2}}{\left(C_{V}T\right)}^{2}]}{\partial x_{i}}+\mu u_{h,i}.$$

When the thermomass theory is applied in the framework of GENERIC, the counterpart equation (Equation (47)) is expressed in the form of thermomass density and thermomass momentum. In order to obtain more obvious comparisons, the same variables are substituted in these equations. They are the energy density *e*(*x*,*t*) and heat flux **q**(*x*,*t*), which are more common in hyperbolic heat conduction equations. Substituting the relationships Equations (4) and (5) into Equations (46) and (47):(49)$$\frac{\partial e}{\partial t}+\frac{\partial q_{i}}{\partial x_{i}}=0,$$
(50)$$\frac{\partial q_{i}}{\partial t}+\frac{\partial (\frac{q_{i}q_{j}}{e})}{\partial x_{j}}=-\varepsilon \frac{e}{c^{2}}\frac{\partial e}{\partial x_{i}}-\kappa_{i}\gamma_{i}q_{i},$$
are derived. The first equation (Equation (49)) is the heat conservation equation. The second equation (Equation (50)) is compared with the momentum equation in thermomass theory (Equation (7)), which is also rewritten in the form of *e* and **q**:(51)$$\frac{\partial q_{i}}{\partial t}+\frac{\partial (\frac{q_{i}q_{j}}{e})}{\partial x_{j}}=-\frac{2\gamma e}{c^{2}\rho }\frac{\partial e}{\partial x_{i}}-\frac{2\gamma eC_{V}}{Kc^{2}}q_{i}.$$

The first term in the left of Equation (51) represents the temporal inertial effect of heat and the second term is the convective term, representing the spatial inertial effect. Hyperbolic heat conduction phenomena arise because of the temporal inertial force of thermomass. Then, the spatial inertial force influences the interactions between the heat fluxes in different directions. The third term in Equation (51) is the pressure gradient, characterized by the gradient of energy density. The last term is the resistance, leading to the decrease of heat flux. In the whole process, the heat is conserved and transferred, while the thermomass momentum is reduced.

By comparing Equation (50) and Equation (51), it is found that they have the same structure and similar terms. The difference is in the definition of the coefficients. In the thermomass momentum equation, the pressure term is derived from the thermal pressure in solid. The resistance is obtained from the analogy to the porous media, and it is assumed that the resistance force has a linear relation with the drift velocity of thermomass gas [33,36]. In the GENERIC equation, the third term, the pressure potential, is obtained from the reversible process, representing the combination of thermomass kinetic energy and thermomass potential energy. Furthermore, the fourth term comes from the dissipation in the irreversible process, leading to entropy production and thermomass energy dissipation [51].

### 4.2. Thermomass Energy

In the GENERIC theory, the reversible process and irreversible process are controlled by two individual potentials. In the reversible process, Hamiltonian potential energy *H*, which consists of thermomass kinetic energy and thermomass potential energy, takes effect, while in the irreversible process, dissipation potential governs and keeps damping monotonously. It was found in this paper that these two potentials have similar forms and they can be unified as one potential energy, namely the thermomass energy.

It was noticed that in the reversible process, thermomass energy is conserved, while in the irreversible process, resistance dissipates the thermomass energy. These behaviors are much like the controlling potentials in GENERIC theory. Therefore, the Hamiltonian potential and dissipation potential in the thermomass system with state variables *ρ_h_* and **m***_h_* can be written as:(52)$$H_{h}=\Xi_{h}=\sum_{i}\frac{1}{2}\kappa_{i}{\left(\eta_{m_{h,i}}\right)}^{2}+\frac{1}{2}\varepsilon \rho_{h}^{2}=\sum_{i}\frac{1}{2}\frac{m_{h,i}^{2}}{\rho_{h}}+\frac{1}{2}\varepsilon \rho_{h}^{2}.$$
The following relationships are required to be satisfied:(53)$$\kappa_{i}\gamma_{i}^{2}=\frac{1}{\rho_{h}},$$
(54)$$\kappa_{i}\gamma_{i}=\frac{1}{\tau_{i}},$$
where *τ* is the relaxation time. In this way, it can be answered what is dissipated in the process in heat conduction. It is the thermomass energy *H*. In the reversible process, thermomass energy is transmitted and conserved, with the transformation between thermomass kinetic energy and thermomass potential energy, while in the irreversible process, thermomass energy is dissipated monotonously. The behaviors of thermomass are much similar with those of entropy. However, the irreversible process leads to the increase of entropy production, while the thermomass energy is reduced due to its dissipation. 

The thermomass energy analysis has two advantages. Firstly, it provides a uniform formulation for the Hamiltonian potential and the dissipation potential. From nonequilibrium state to equilibrium state, the thermomass energy keeps being dissipated monotonously, which can be used as the Lyapunov function. Secondly, it gives a clearer and more intrinsic physical picture for the heat transport in GENERIC theory, where reversible process is described in energy representation while the irreversible one is in entropy representation. 

### 4.3. Extension in the Framework of GENERIC

The reduction from microscale view to macroscale view ignores some details and focuses only on the variation of macroscopic patterns. Nevertheless, different levels of descriptions are required in different situations. For example, in an equilibrium system, only one state variable is necessary, namely the total thermal energy *E*. However, in a system described by Fourier’s law, the distribution of local energy density *e*(*x*) is supposed to be considered. If more proper variables are added to the list, more details of the system are revealed. However, the word *proper* is significant, which requires to pick up the relatively more general variables and leave out the less significant ones and that the state variables are rational. For example, when both heat flux and the gradient of heat flux influence the heat conduction process, the heat flux is obviously more general, while the gradient of it might be only a good supplement.

One advantage of the GENERIC framework is that it is easy to extend because it provides the mathematical structure. Thus, flexible state variables can be chosen according to different systems. Of course, the most important thing is that these state variables are reasonable for the target system. One possible extension is the distortion matrix, in which case the propagation of heat suffers from viscosity, and heat fluxes in different directions have stronger interactions with each other. 

At first, the concept of volume density of labels **a** is introduced [51,56]. In Lagranian mechanics, the coordinate is set to be with the thermomass particles and the particle trajectories are recorded. However, when these equations are transformed into Euler representation, the coordinates might be compressed or twisted. The volume density of labels **a**, which is a vector, implies where the particles come from. The gradient of **a**, called distortion matrix **D**, shows the torsion and deformation of the given volume, which has been applied in elasto-plastic solid successfully [56]. The distortion matrix **D** is usually used to describe the solid-like fluid or fluid-like solid, which has the flowing and elasticity characteristics at the same time. In heat conduction, even though the thermomass gas is regarded as a kind of fluid, the real transport process is complex and these kinds of fluids might have the solid characteristics. Therefore, it is reasonable to apply the theory for elasto-plastic solid to thermomass gas. Actually, the historical effect of heat flux with temperature [7] or the vibration interference [57], for example, can be seen as some kind of elasticity. It is thought that the distortion matrix is relevant with the initial state of thermal energy distribution and represents the viscous effect and elastic effect. 

The time evolution equations with label field **a** are derived as the first step. The derivatives of label field **a** with respect to displacement and momentum are defined as:(55)$$\frac{\partial a_{i}}{\partial x_{j}}=D_{ij},$$
(56)$$\frac{\partial a_{i}}{\partial p_{j}}=0.$$
Thus, the Poisson bracket, consisting of thermomass density *ρ_h_*, thermomass momentum **m***_h_* and label field **a** has the form of:(57)$$\begin{array}{ll} {\left\{A,H\right\}}^{label} & ={\left\{A,H\right\}}^{c}+{\left\{A,H\right\}}^{a}\\ & =\int_{V}\rho_{h}(\frac{\partial A_{\rho_{h}}}{\partial x_{i}}H_{m_{h,i}}-\frac{\partial H_{\rho_{h}}}{\partial x_{i}}A_{m_{h,i}})dr\\ & +\int_{V}m_{h,i}(\frac{\partial A_{m_{h,i}}}{\partial x_{j}}H_{m_{h,j}}-\frac{\partial H_{m_{h,i}}}{\partial x_{j}}A_{m_{h,j}})dr\\ & +\int a_{i}(\frac{\partial A_{a_{i}}}{\partial x_{j}}H_{m_{h,j}}-\frac{\partial H_{a_{i}}}{\partial x_{j}}A_{m_{h,j}})dr \end{array}$$
Via integration by parts, the time evolution of these state variables yields:(58)$$\frac{\partial \rho_{h}}{\partial t}+\frac{\partial }{\partial x_{i}}(\rho_{h}H_{m_{h,i}})=0,$$
(59)$$\frac{\partial a_{i}}{\partial t}+\frac{\partial (a_{i}H_{m_{h,j}})}{\partial x_{j}}=0,$$
(60)$$\frac{\partial m_{h,i}}{\partial t}+\frac{\partial }{\partial x_{j}}(m_{h,i}H_{m_{h,j}})=\frac{\partial \pi^{†}}{\partial x_{i}},$$
(61)$$\pi^{†}=H-\rho_{h}H_{\rho_{h}}-m_{h,j}H_{m_{h,j}}-a_{j}H_{a_{j}}.$$
From the equations above, it is found that label field takes part in a pure convective process in reversible systems. The force potential $\pi^{†}$ differs from that in Equation (24) by one energy term produced by label vector. Furthermore, if the distortion matrix is taken as the state variable and the Hamiltonian energy *H* is a function of only the distortion matrix, not the function of label field, the Poisson bracket (57) is transformed into [56]:(62)$$\begin{array}{ll} \dot{A} & ={\left\{A,H\right\}}^{c}+{\left\{A,H\right\}}^{D}\\ & =\int_{V}\rho_{h}(\frac{\partial A_{\rho_{h}}}{\partial x_{i}}H_{m_{h,i}}-\frac{\partial H_{\rho_{h}}}{\partial x_{i}}A_{m_{h,i}})dr\\ & +\int_{V}m_{h,i}(\frac{\partial A_{m_{h,i}}}{\partial x_{j}}H_{m_{h,j}}-\frac{\partial H_{m_{h,i}}}{\partial x_{j}}A_{m_{h,j}})dr\\ & +\int_{V}D_{ij}(H_{m_{h,j}}\frac{\partial A_{D_{ik}}}{\partial x_{k}}-A_{m_{h,j}}\frac{\partial H_{D_{ik}}}{\partial x_{k}})dr\\ & +\int_{V}(\frac{\partial D_{ij}}{\partial x_{k}}-\frac{\partial D_{ik}}{\partial x_{j}})(A_{D_{ik}}H_{m_{h,j}}-H_{D_{ik}}A_{m_{h,j}})dr \end{array}$$

However, the irreversible process might destroy the continuity of label field **a** and Equation (55) might not be always satisfied, which results in the failure of Jacobi identity. The last part in Equation (62) is added to the equation to make sure that it fulfills the Jacobi identity even for incompatible distortion fields. The time evolution equations derived are:(63)$$\frac{\partial \rho_{h}}{\partial t}+\frac{\partial }{\partial x_{i}}(\rho_{h}H_{m_{h,i}})=0,$$
(64)$$\frac{\partial D_{ij}}{\partial t}+\frac{\partial (D_{ik}H_{m_{h,k}})}{\partial x_{j}}-(\frac{\partial D_{ik}}{\partial x_{j}}-\frac{\partial D_{ij}}{\partial x_{k}})H_{m_{h,k}}=0,$$
(65)$$\frac{\partial m_{h,i}}{\partial t}+\frac{\partial }{\partial x_{j}}(m_{h,i}H_{m_{h,j}})=-\rho_{h}\frac{\partial H_{\rho_{h}}}{\partial x_{i}}-m_{h,j}\frac{\partial H_{m_{h,j}}}{\partial x_{i}}-\frac{\partial (D_{ji}H_{D_{jk}})}{\partial x_{k}}+\frac{\partial D_{jk}}{\partial x_{i}}H_{D_{jk}}.$$

Due to the complex distortion matrix, a universal form of force potential is difficult to find. A possible form of *H* is:(66)$$H_{2}^{D}=\sum_{i}\frac{1}{2}\frac{m_{h,i}^{2}(x,t)}{\rho_{h}(x,t)}+\frac{1}{2}\varepsilon \rho_{h}^{2}(x,t)+\sum_{i,j}\frac{1}{2}\chi_{ij}D_{ij}^{2}.$$

In fact, Equation (65) implies that the heat flux is not only determined by the gradient along this direction, but also interacts with the gradients of variables along other directions. The physical meaning of the distortion matrix, the elasticity of thermomass, can be regarded as a nonlocal effect, because it takes into consideration the interactions between the thermomass particles and their surroundings. Its formulation is similar with that of Rational Extended Thermodynamics [47]. However, the closure problems are avoided and it closes the equation set by the application of dissipation potential, as shown in the following part.

The new variables also have their own part in the dissipation process, which is irreversible. When the phonon entropy is the function of label field **a** or distortion matrix field **D**, or even the gradient of the distortion matrix, which is a third-order tensor, the dissipation terms have different characteristics. Here, for simplicity, it is assumed that the eta-function is the function of the distortion matrix **D**. Then the time evolution Equation (64) is rewritten as:(67)$$\frac{\partial D_{ij}}{\partial t}+\frac{\partial (D_{ik}H_{p_{h,k}})}{\partial x_{j}}-(\frac{\partial D_{ik}}{\partial x_{j}}-\frac{\partial D_{ij}}{\partial x_{k}})H_{p_{h,k}}=\frac{\partial \Xi }{\partial \eta_{D_{ij}}}.$$

One possible example of Ξ and *η* is proposed. The dissipation potential is set to be a quadratic function, as:(68)$$\Xi^{D}=\frac{1}{2}\beta_{ij}{\left(\eta_{D_{ij}}\right)}^{2}.$$
The irreversible process requires that the dissipation potential always keeps damping during the process of propagation, thus, the terms is supposed to give the equation the tendency to equilibrium. Analogous with the behavior of heat flux, the relationship between eta-function and distortion matrix field is set to be:(69)$$\eta^{D}=\eta_{0}-\frac{\varphi_{ij}}{2}{\left(D_{ij}\right)}^{2},$$
In the above equations, *β* and φ are the coefficients related to the material thermal properties. The complete time evolution equation for **D** is then rewritten as:(70)$$\frac{\partial D_{ij}}{\partial t}+\frac{\partial (D_{ik}H_{p_{h,k}})}{\partial x_{j}}-(\frac{\partial D_{ik}}{\partial x_{j}}-\frac{\partial D_{ij}}{\partial x_{k}})H_{p_{h,k}}=-\beta_{ij}\varphi_{ij}D_{ij},$$

The state variables that are necessary to describe the system are changing in different problems and at different description levels. Taking more state variables does not always mean better, because while some patterns are negligible, more details can mean more meaningless efforts and disordered importance. In view of the current experimental measurement techniques, label field and distortion matrix field are both difficult to observe. However their effects are presented by the thermomass density *ρ_h_* and thermomass momentum **m***_h_*. This article makes an effort to demonstrate that the heat conduction principles for thermomass system might be more complex than was thought and the theory can be developed to a more general stage.

Inner heat source absorbs and emits heat, providing a new term to the dissipation potential. Different from distortion matrix, inner heat source is not a state variable, but a given system parameter. In a system represented by the state variables *ρ_h_* and **m***_h_*, the inner heat source has nothing to do with Hamiltonian mechanics and it influences the heat conduction process by changing the dissipation potential. The time evolution equations of thermomass system with inner heat source are supposed to be:(71)$$\frac{\partial \rho_{h}}{\partial t}+\frac{\partial }{\partial x_{i}}(\rho_{h}H_{m_{h,i}})=s_{h},$$
(72)$$\frac{\partial m_{h,i}}{\partial t}+\frac{\partial }{\partial x_{j}}(m_{h,i}H_{m_{h,j}})=\frac{\partial \pi }{\partial x_{i}},$$
according to the physical interpretation, where *s_h_* is the inner thermomass source, representing the mass density variation per unit volume per second. Therefore, to make new potential quantities compatible with Equations (71) and (72), the Hamiltonian potential and dissipation potential are changed into:(73)$$H_{2,inner}=\sum_{i}\frac{1}{2}\frac{m_{h,i}^{2}(x,t)}{\rho_{h}(x,t)}+\frac{1}{2}\varepsilon_{2}\rho_{h}^{2}(x,t),$$
(74)$$\Xi =\frac{1}{2}\sum_{i}\kappa_{i}\eta_{p_{h,i}}^{2}+s_{h}\eta_{\rho_{h}}=\frac{1}{2}\sum_{i}\kappa_{i}\gamma_{i}^{2}m_{h,i}^{2}+s_{h}\eta_{\rho_{h}}.$$
The analyses of dissipation potential show that the thermomass source potential is written as the product of the derivative of eta-function with respect to the thermomass density *ρ_h_* and the inner thermomass source *s_h_*. If this inner source is a hot source *s_h_* > 0, it adds to the dissipation potential. Otherwise, it reduces the dissipation potential. 

Distortion matrix and inner heat sources are used as the examples to demonstrate how to extend the thermomass theory in the framework of GENERIC. The state variables are chosen according to the characteristics of the system. Inner heat sources gives a new term to the dissipation potential, and the distortion matrix allows for stronger interactions between heat fluxes in different directions, describing the viscous effect and elasticity of thermomass gas.

## 5. Numerical Results

### 5.1. Influences of the Convective Term

One of the characteristics of Equation (50), the time evolution equation for the thermomass momentum, is the second term which implies the momentum convection effects. The influences of the convective term on the propagation behaviors are investigated in this section by numerical simulations. Here, the heat conduction problems are solved in one-dimensional regimes and two-dimensional regimes.

The transient heat conduction phenomenon in a large plate with finite length as shown in Figure 1, is studied, which can be regarded as a one-dimensional heat conduction problem. The left boundary is heated by a sine-shape heat pulse for the time from 0 to *t_pulse_*:
(75)$$q_{x}=\left\{ \begin{array}{ll} Y_{1}(1-\cos(2\pi t/t_{pulse}) & 0<t<t_{pulse}\\ 0 & t>t_{pulse} \end{array}\right..$$
where *Y*_1_ is the amplitude of heat pulse. Before calculations, the state variables and coefficients are normalized and the numerical algorithm comes from computational fluid mechanics [58,59,60]. After normalization, *Y*_1_ is chosen to be 1. More numerical details can be seen in the Appendix A. If the thermal conductivity is constant, the effects of the convective term can be highlighted. Thus, in Figure 2 the problem with constant thermal conductivity is analyzed. That is, the equation:(76)$$\frac{\partial q}{\partial t}+\frac{\partial (q\frac{q}{e})}{\partial x}=-k\frac{\partial e}{\partial x}-\frac{1}{\tau }q,$$
is solved, where the second term denotes the convective effect. In the simulations, *k* and *τ* are both set to be 1 to make the calculations easier. The choices of these coefficients are important, of course. However, it does not influence the qualitative characteristics of the equations, as it has been adopted in the reference [59,60,61,62]. The propagation patterns of the energy density with and without the second term are displayed in Figure 2. The results for the equations without the second term are plotted in solid line while the dash line denotes those for the equations with convective term. It can be seen that the heat pulse at the left boundary produces a thermal wave in the medium and it keeps the original pattern during propagation while the peak value is dissipated as shown by the solid line. When the convective term is considered, the shape is changed and a sharp wave front comes into being. The convective term shows the drift behaviors of thermomass conveyed by momentum. Thus, it is an obvious deduction when the mass nature of heat is taken into consideration. Meanwhile, the existence of convective term speeds up the thermal wave and leads to higher dissipation rate for the peak value of the patterns.

To show more characteristics of the momentum convection, a two-dimensional heat conduction problem is considered where heat transport occurs in a plate with finite length and width and infinite height. Then, the left boundary is heated by a partial heat pulse:(77)$$q_{x}=\left\{ \begin{array}{ll} Y_{2}(1-\cos(2\pi t/t_{pulse})(1-\cos[4\pi (y-0.25)]) & 0<t<t_{pulse},0.25<y<0.75\\ 0 & t>t_{pulse}\;\mathrm{or}\;y>0.75\;\mathrm{or}\;y<0.25 \end{array}\right.,$$
where *Y*_2_ is also set to be 1. If it is not clarified specifically, all the coefficients are set to be 1. The distribution shape of the heat flux is displayed in Figure 3. Then, the energy distribution along with time is shown in Figure 4 and different colors denotes different values of the energy density, which has a corresponding relationship with temperature. Figure 4a–c are the energy distribution propagation patterns *e*_n_ for Equations (49) and (50) without the convective term and Figure 4d–f are those *e*_c_ with the convective term. The thermal energy enters the regime from the left boundary and it spreads away just like water ripples. The majority of the energy propagates forward, namely along with the original direction of the heat flux, while the nonequilibrium energy distribution produces the heat flux in other directions. The difference of the energy distribution (*e*_c_-*e*_n_) is plotted in Figure 5. It was found that the convective term enhances the spreading behaviors of the thermal energy. In macroscale descriptions, the spreading enhancement caused by the convective term is similar with the viscous effect. However, they are different because the convective effect is reversible and the viscous effect is irreversible.

### 5.2. Influences of the Distortion Matrix

Distortion matrix in thermomass theory, analogous to that in solid mechanics, implies the concept of thermomass elasticity. For example:(78)$$D_{xx}=\frac{\partial a_{x}}{\partial x},$$
means the compression or tension of the label field and:(79)$$D_{xy}=\frac{\partial a_{x}}{\partial y},$$
implies the torsion of the label field. Assuming that the Hamiltonian potential is the function of the distortion matrix:(80)$$H_{2d}^{D}=\frac{1}{2}\varepsilon \rho_{h}^{2}+\frac{1}{2}\frac{m_{h,x}^{2}}{\rho_{h}}+\frac{1}{2}\frac{m_{h,y}^{2}}{\rho_{h}}+\frac{1}{2}\chi_{xx}D_{xx}^{2}+\frac{1}{2}\chi_{yy}D_{yy}^{2},$$
then the heat conduction is also influenced by the distortion matrix coefficients *D_xx_* and *D_yy_*, which reveals a tendency of recovering to the initial state. In the numerical simulations, the nondimensional coefficients *χ_xx_*, *χ_yy_* are set to be 0.5 and 0.5, and other coefficients are 1. 

The energy distribution *e*_D_ at time *t* = 0.3, which is derived from the equation set with Hamiltonian energy (Equation (76), is plotted in Figure 6a and the energy distribution *e*_c_ which is another view of Figure 4e is displayed in Figure 6b. These two figures are compared to highlight the influences of the distortion matrix. Moreover, their difference, *e*_D_-*e*_c_, is shown in Figure 7. The distortion coefficients *D_xx_* and *D_yy_* speeds up the wave propagation and the wave shape are also altered. Meanwhile, the coefficients enhance the fluctuations of wave. In Figure 7, another small wave peak appears behind the first wave pattern, and in some sense, it shows the effect of elasticity. Considering that the distortion matrix field goes through an irreversible process at the same time, the introduction of these coefficients will add to the dissipation and strengthen the viscous effects. One obvious feature of the viscous effects is that the peak value of the energy density in Figure 6a is lower than that in Figure 6b.

## 6. Conclusions

The thermomass theory and GENERIC theory deal with the non-Fourier heat conduction phenomena in different ways. The core idea of the thermomass theory is to reveal the mass nature of heat during heat conduction based on Einstein’s mass-energy relation. Therefore, heat transfer is actually the flow of thermomass gas in porous media, thus, the principle of fluid mechanics can be used to describe the heat conduction process in the media. The momentum conservation equation of thermomass gas is actually a wave equation with thermal inertia force and damping. The GENERIC theory constructs an abstract geometrical structure for various transport processes. Its core idea is to adopt the contact geometry to derive the time evolution equations of the state variables, by handling the reversible and irreversible process separately. The reversible process is expressed in energy representation governed by Hamiltonian energy, while the irreversible process is expressed in entropy representation dominated by dissipation potential. 

Analyses show that the governing equations of heat conduction derived from GENERIC theory are the same as that derived from thermomass theory, if proper coefficients and potentials are chosen. The reversible process in GENERIC theory corresponds to conversion process between the thermomass potential energy and the thermomass kinetic energy in thermomass theory, while the irreversible process in GENERIC theory, which is described by dissipation potential, corresponds to the dissipation process of thermomass energy due to the resistance in thermomass theory. GENERIC theory provides a concrete mathematical foundation for the thermomass theory, while thermomass theory gives GENERIC more physical pictures. All these indicate that both GENERIC theory and thermomass theory are not only compatible, but also supportive and complementary mutually.

The advantage of thermomass theory is that it not only reveals the physical meanings of the Hamiltonian energy and the dissipation potential in GENERIC, but also uniformly expresses these two kinds of energy as one unified quantity. Hamiltonian energy in the reversible process corresponds to the conversion process of potential energy and kinetic energy of thermomass during wave transport of heat (thermomass), and the dissipation potential in the irreversible process corresponds to the dissipation process of thermomass energy. One advantage of GENERIC theory is that it provides a flexible structure, which is easy to include more new state variables, such as the inner heat source and the distortion matrix field. Numerical methods are adopted to investigate the influences of the convective term and distortion matrix term on the wave propagation patterns. The convective term establishes a sharp wave front and enhances the spreading of thermal energy. Thus, it leads to more dispersive energy distribution. The distortion matrix field considers the viscosity and elasticity of thermomass gas and constructs a new series of heat conduction equations, as is shown in Section 5. The application of these new state variables enables thermomass theory to describe more complex heat conduction phenomena.

## Figures and Tables

**Figure 1 entropy-22-00227-f001:**
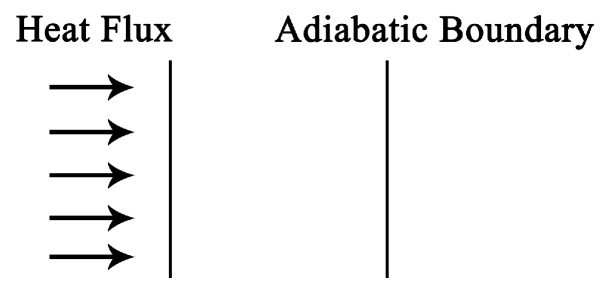
Schematic diagram of the numerical regime for one-dimensional heat conduction problem.

**Figure 2 entropy-22-00227-f002:**
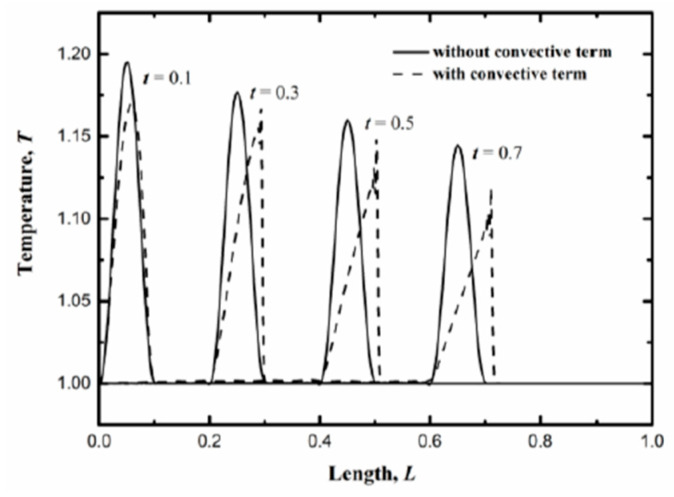
Temperature distributions of Equation (76) at time *t* = 0.1, 0.3, 0.5 and 0.7. The solid line denotes the results without the convective term and the dash line denotes the results with the convective term.

**Figure 3 entropy-22-00227-f003:**
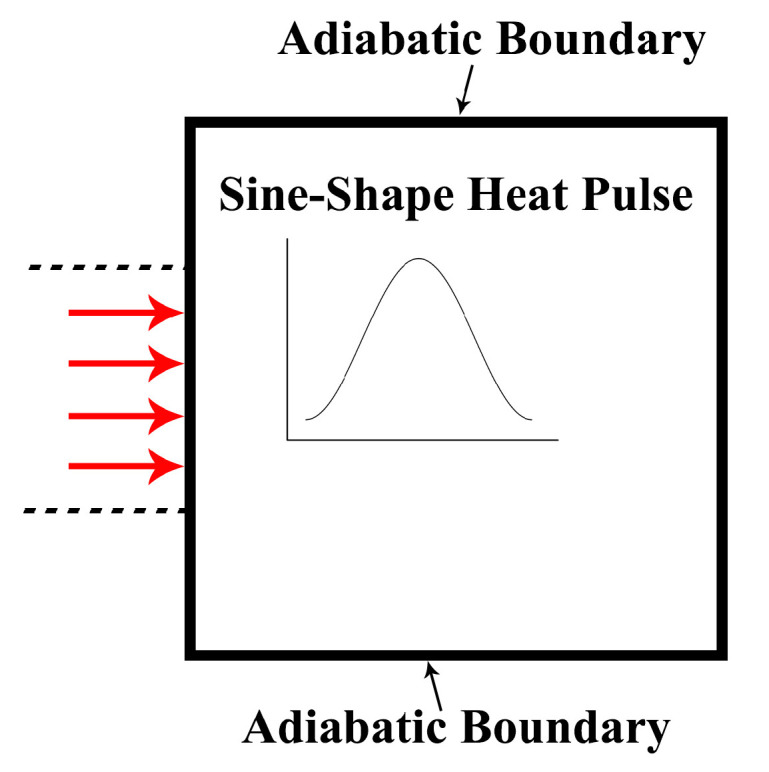
Schematic diagram of the numerical regime for the two-dimensional heat conduction problem.

**Figure 4 entropy-22-00227-f004:**
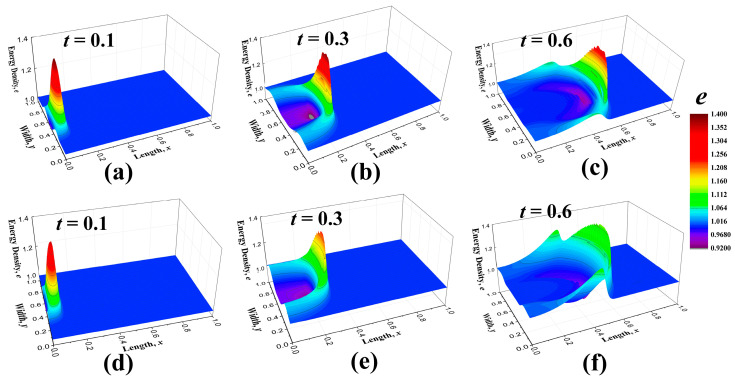
Energy density distribution patterns at time *t* = 0.1, 0.3 and 0.6. (**a**–**c**) show the figures *e*_n_ obtained from the equations without heat flux convective term, while (**d**–**f**) are the figures of the results *e*_c_ for the equations with heat flux convective term. The different color denotes the values of energy density *e*_n_ and *e*_c_.

**Figure 5 entropy-22-00227-f005:**
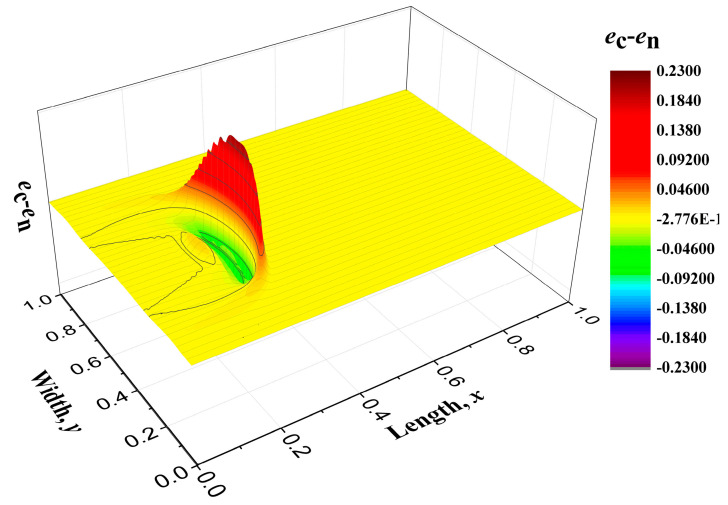
The energy difference patterns *e*_c_-*e*_n_ at time *t* = 0.3 between those with a convective term and those without a convective term.

**Figure 6 entropy-22-00227-f006:**
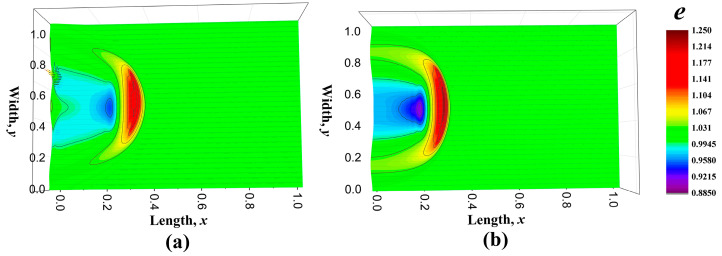
Energy distribution patterns at time *t* = 0.3 for the equations with distortion matrix and convective term *e*_D_ (**a**) and those equations without distortion matrix *e*_c_ (**b**).

**Figure 7 entropy-22-00227-f007:**
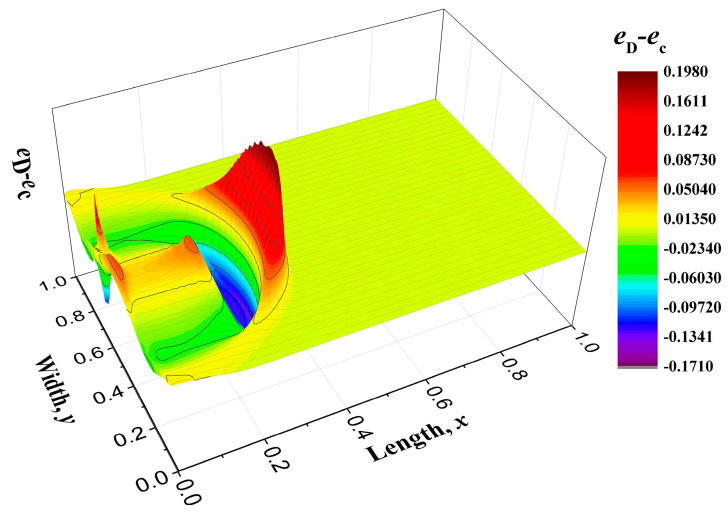
Energy difference patterns *e*_D_-*e*_c_ between those with distortion matrix *e*_D_ and those without distortion matrix *e*_c_ when the Hamiltonian energy *H* is the function of D*_xx_* and D*_yy_*.

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
