# Peer review of "Thermomass Theory in the Framework of GENERIC"

_entropy, 2020, doi:10.3390/e22020227_

Round 1

Reviewer 1 Report

Nanotechnologies have actively contributed to main advances in modern computers and electronics. Their correct use requires, however, to reinvestigate some concepts which are well-established at macroscale, but fail when the characteristic sizes go down to the nanoscale wherein nonlocal, nonlinear and memory effects are important. For example, the classical Fourier law, which can be always used in common applications at macroscale to relate the local heat flux to the gradient of temperature by means of the thermal conductivity, is no longer adequate. As a consequence, several empirical and theoretical models beyond Fourier law have been introduced in last decades. Among those models, that based on the Thermomass Theory is surely worth of consideration. In this paper, the heat-transport equation of the Thermomass Theory is firstly compared with that of arising from the GENERIC and than applied to study the propagation of heat waves. In my opinion the article can be accepted as it is.

A small final comment: Equation 14 is able to take into account memory, nonlocal and nonlinear effects, as well as other theoretical models. On the propagation of heat waves in two-dimensional for example, it may be interesting to compare the results arising from the different models with those arising from Eq. (14). Since heat waves can be experimentally observed (or generated), they can be a possible benchmark for the different theoretical models.  

Author Response

Nanotechnologies have actively contributed to main advances in modern computers and electronics. Their correct use requires, however, to reinvestigate some concepts which are well-established at macroscale, but fail when the characteristic sizes go down to the nanoscale wherein nonlocal, nonlinear and memory effects are important. For example, the classical Fourier law, which can be always used in common applications at macroscale to relate the local heat flux to the gradient of temperature by means of the thermal conductivity, is no longer adequate. As a consequence, several empirical and theoretical models beyond Fourier law have been introduced in last decades. Among those models, that based on the Thermomass Theory is surely worth of consideration. In this paper, the heat-transport equation of the Thermomass Theory is firstly compared with that of arising from the GENERIC and than applied to study the propagation of heat waves. In my opinion the article can be accepted as it is.

Reply:

Thank you very much for your comments. We have revised the manuscript.

A small final comment: Equation 14 is able to take into account memory, nonlocal and nonlinear effects, as well as other theoretical models. On the propagation of heat waves in two-dimensional for example, it may be interesting to compare the results arising from the different models with those arising from Eq. (14). Since heat waves can be experimentally observed (or generated), they can be a possible benchmark for the different theoretical models.

Reply:

Thank you very much. There are numerous theoretical models for non-Fourier heat conduction phenomenon. However, the actual heat transport process is more complex. For example, the current models can’t describe the spatial dispersion effect. Different models solved the problems from different ways and can be applied in different materials. As for different cases, the suitable models might be different.

It is interesting to compare different models in two-dimensional regime, since thermal waves have been observed in graphite recently. Besides, different terms are included in different models and they could lead to different phenomena. In this paper, we discussed the effects of convective term and distortion term. In future, we could explore the effects of more terms.

Reviewer 2 Report

Please find my review enclosed.

The Authors are trying to do their best, and some small improvements are visible. However, it still consists of many ambiguities, especially about the basic concepts and structure of thermomass equations. 

In order to find some solution, I have recommended to include remarks about the differences between the usual continuum theories and their approach. They should call the reader's attention to the questions. 

Moreover, the new section about the numerical solution is too weak to accept for publication. In this regards, I have recommended its complete revision due to the following reasons:
1, the boundary conditions are discussed adequately,
2, the discretization is also not clear and there are unclear parts, too;
3, there is no reference here, but definitely, they are not the first who applies staggered fields for heat equations;
4, the quality of the figures is too low, especially their resolution.

Now, I am concluding at major revision. That could be enough to make the paper acceptable. 

Best regards,

Round 2

Reviewer 2 Report

I accept the modifications performed in the manuscript.

However, the Reviewer would like to express two general comments regarding the answers:

According to my knowledge, it is not physically consistent to arbitrarily choose to include relativistic effects without proper space-time and relativistic formulation. That part is completely missing in the framework of thermomass theories.  Normalization is one thing. Introducing dimensionless parameters is indeed a common method to fit together the time and space scales. However, it does not mean that any other parameter is transformed or set to unity. In the Ref. you mentioning the unphysical behavior of MCV equation, it is not possible to adjust the propagation speed (which is quite weird in some sense).

Anyway, I suggest only one minor correction: the equation S.12 misses the derivative of "e" (as it is also a field variable). 

Author Response

This manuscript is a resubmission of an earlier submission. The following is a list of the peer review reports and author responses from that submission.

Round 1

Reviewer 1 Report

Please, find attached the pdf document containing the opinion of the Reviewer.

Reviewer 2 Report

This new version of the paper displays a better layout than the previous one, i.e., in the present version of the paper one can clearly see which is the real goal. The numerical part might gave novelties to the paper; nevertheless, in this new version it has been deleted. In the opinion of this referee, this yields that the new results are quite poor. As a consequence, the paper does not seem to be suitable for acceptance.